# WildBench: Benchmarking LLMs with Challenging Tasks from Real Users in the Wild

**Bill Yuchen Lin**[♡◇]

**Yuntian Deng**[♡]    **Khyathi Chandu**[♡]    **Faeze Brahman**[♡]    **Abhilasha Ravichander**[♡]
**Valentina Pyatkin**[♡]    **Nouha Dziri**[♡]    **Ronan Le Bras**[♡]    **Yejin Choi**[♡◇]

[♡]Allen Institute for AI    [◇]University of Washington
🤗 https://hf.co/spaces/allenai/WildBench

## Abstract

We introduce WildBench, an automated evaluation framework designed to benchmark large language models (LLMs) using challenging, real-world user queries. WILDBENCH consists of 1,024 examples carefully selected from over one million human-chatbot conversation logs. For automated evaluation with WILDBENCH, we have developed two metrics, WB-Reward and WB-Score, which are computable using advanced LLMs such as GPT-4-turbo. WILDBENCH evaluation uses task-specific checklists to evaluate model outputs systematically and provides structured explanations that justify the scores and comparisons, resulting in more reliable and interpretable automatic judgments. WB-Reward employs fine-grained pairwise comparisons between model responses, generating five potential outcomes: much better, slightly better, slightly worse, much worse, or a tie. Unlike previous evaluations that employed a single baseline model, we selected three baseline models at varying performance levels to ensure a comprehensive pairwise evaluation. Additionally, we propose a simple method to mitigate length bias by converting outcomes of "slightly better/worse" to "tie" if the winner's response exceeds the loser's by more than $K$ characters. WB-Score evaluates the quality of model outputs individually, making it a fast and cost-efficient evaluation metric. WILD-BENCH results demonstrate a strong correlation with the human-voted Elo ratings from Chatbot Arena on hard tasks. Specifically, WB-Reward achieves a Pearson correlation of 0.98 with top-ranking models. Additionally, WB-Score reaches 0.95, surpassing both ArenaHard's 0.91 and AlpacaEval2.0's 0.89 for length-controlled win rates, as well as the 0.87 for regular win rates.

## 1 Introduction

Large language models (LLMs) have become integral to a wide range of real-world applications due to their strong generalization capabilities across diverse tasks. However, effectively evaluating their performance remains a challenging problem, particularly when striving for an automated and cost-effective solution. Traditional benchmarking datasets like MMLU (Li et al., 2023a) focus primarily on assessing the reasoning abilities of LLMs using multiple-choice questions, which fall short in evaluating the more open-ended problems that real-world users pose. Chatbot Arena (Chiang et al., 2024) provides an online platform where human preferences are collected to judge pairs of model outputs, subsequently ranking LLMs using Elo ratings. While this human-based evaluation method offers valuable insights into user preferences, it has notable limitations, such as high labor costs, the inability to deliver real-time results, a lack of data transparency, and the challenge of fairly evaluating all models with the same data.

Several automated benchmarks such as AlpacaEval (Li et al., 2023b), MT-bench (Zheng et al., 2024), and ArenaHard (Li et al., 2024) employ advanced LLMs like GPT-4-Turbo to assess the quality of model responses. Comparative analyses of these benchmarks are presented in Table 1 and Figure 3. These existing benchmarks exhibit significant shortcomings in task composition and skill coverage, particularly in mirroring the natural distribution of real-world user tasks. MT-bench, comprising

| What is the capital of Australia?
What is some cool music from the 1920s?
How do I wrap a present neatly?
Can you write code?
~20 recipe generation tasks   **AlpacaEval** | Please provide me python code to go through a directory and its subdirectories and delete images that are not horizontal. |
|---|---|

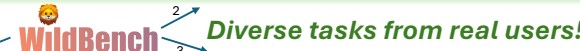

*Diverse tasks from real users!*

| hey can you write an essay on the impact of the G20 summit on the global economy, trade, development and the role of young people in shaping the future of the world, it has to have more than 1200 words. Write it beautiful and poetic. Use extensive vocabulary. Use a lot of factual and empirical data. Use some, ancient indian historical references. | I want to create an open source, highly realistic and grounded text-based business simulation game that is played in the terminal, with a large range of different features that make the game as realistic a simulation as possible. In light of this the game should not have set values for anything because that is unrealistic - real life isn't like that; the sim should be as close to reality as possible. I will host it on Github. Please create a FULL, COMPLETE file structure for the game's Github repo. |
|---|---|

Figure 1: Example tasks sampled from AlpacaEval (Li et al., 2023b) and WILDBENCH. Tasks in WILDBENCH are more diverse and challenging, which are collected from real users in the wild. Complex real-user tasks usually have multiple constraints and require higher-order reasoning skills, which are well represented in WILDBENCH.

only 80 hand-crafted examples, lacks sufficient breadth for a comprehensive evaluation. Meanwhile, AlpacaEval, with 805 tasks derived from multiple alignment datasets, includes relatively simple tasks, such as "*What is the capital of Australia?*" and suffers from low task diversity; for instance, over 20 tasks redundantly assess recipe generation skills (e.g., "can you provide a recipe for ...?"). We show a few examples in Figure 1 to illustrate the differences between AlpacaEval and our WILDBENCH.

AlpacaEval mostly focuses on information-seeking tasks, containing merely 6% coding and 3% mathematics tasks. Conversely, ArenaHard, sampling 500 tasks from ChatbotArena, displays an excessive concentration on coding and debugging tasks, accounting for over 57% of its content. Most existing benchmarks do not sufficiently challenge the models with the varied and unexpected nature of user inquiries in practical settings, thus limiting their overall effectiveness in providing a holistic evaluation. This issue highlights the necessity for more comprehensive benchmarks that can better simulate the wide range of tasks from real users.

In this paper, we introduce WILDBENCH, an automated evaluation framework designed for assessing LLMs using complex tasks from real-world users. The examples in WILDBENCH are periodically updated, with the current version (V2) comprising 1,024 tasks carefully curated from real user-chatbot dialogs provided by the AI2's WildChat project (Zhao et al., 2024). We engage multiple advanced LLMs to process a filtered selection from WildChat, tasking them with the analysis of the requisite knowledge and skills for each task and subsequently labeling the difficulty level. Tasks considered as easy by all models are excluded. We ensure the distribution of tasks mirrors the original WildChat data, such that the task distribution of WILDBENCH is still natural (Figure 3). Additionally, all finalized tasks undergo manual review. Further details are provided in Section 2.

As shown in Figure 1, WILDBENCH presents a significantly harder challenge due to the complexity, depth, and realism of the tasks involved. WILDBENCH is sourced from real-world user interactions and has been carefully curated to ensure diversity and challenge. The tasks in WILDBENCH typically demand higher-order reasoning, such as writing and/or debugging code with specific constraints, creative writing with multiple constraints on the style and content, or designing a software system with complex requirements. These tasks often require critical thinking, creativity, and technical expertise, making WILDBENCH substantially more challenging than AlpacaEval, where simpler, factual, or surface-level tasks dominate.

WILDBENCH evaluation is illustrated in Figure 4. To design a reliable automatic evaluation, we employ two key designs for using LLMs as judges. Drawing inspiration from how humans evaluate responses to open-ended questions, we develop task-specific checklists. These checklists guide LLMs in generating consistent and reliable judgments, with each checklist comprising questions focused on specific criteria. Similar to the zero-shot Chain-of-Thoughts (CoT) prompting (Kojima et al., 2022), we prompt LLMs to provide step-by-step, structured analyses of each LLM response. This method encourages a detailed, fine-grained evaluation process, culminating in a well-justified final decision.

We employ two primary metrics: WB-Reward for *pairwise* comparisons and WB-Score for *individual* scoring. WB-Reward is based on pairwise comparisons between LLMs, with five possible outcomes: "A is much/slightly better/worse than B" or "Tie." Notably, we used three baseline models to compare with each testing model instead of using a single baseline model, as most prior works do. This approach provides a more comprehensive assessment based on different levels of model performance.

WB-Score measures the quality of each model's generation individually, offering a quicker and more cost-effective evaluation. To mitigate the bias towards longer outputs, a common issue in LLM-as-a-judge evaluations (Dubois et al., 2024), we introduced a simple length-penalty method, converting slight wins/losses to ties when the winner's output is significantly longer than the loser's.

Both metrics have demonstrated strong correlations with human judgments, evidenced by a Pearson correlation of 0.98 for WB-Reward and 0.95 for WB-Score against the human-voted Elo rating from Chatbot Arena on the top-ranking models. These scores significantly surpass other benchmarks, such as ArenaHard(Li et al., 2024)'s 0.91 and AlpacaEval2.0's 0.87 (0.89 for the length-controlled version) (Li et al., 2023b; Dubois et al., 2024), validating WILDBENCH's effectiveness and alignment with human-based evaluation. More details are shown in Table 3 in Section 4.

## 2 WILDBENCH DATA CURATION

In this section, we describe the data curation process for the tasks used to evaluate LLMs in WILD-BENCH . Our goal is to ensure that the selected tasks not only represent real-world use cases but are also challenging enough to distinguish the varying capabilities of LLMs.

Table 1: Statistical comparison of LLM alignment benchmarks. Length are in characters.

| Dataset | #Tasks | #Turns | ChatHistory | QueryLen | PromptLen | RealUser | TaskTag | Evaluation |
|---|---|---|---|---|---|---|---|---|
| **MT-Bench** | 80 | 2 | ✔Dynamic | 202.2 | Dynamic | ✘ | ✔ | Score |
| **AlpacaEval** | 805 | 1 | ✘ | 164.9 | 164.9 | ✘ | ✘ | Pair (ref=1) |
| **ArenaHard** | 500 | 1 | ✘ | 406.4 | 406.4 | ✔ | ✘ | Pair (ref=1) |
| **WILDBENCH** | 1,024 | ≤5 | ✔Static | 978.5 | 3402.1 | ✔✔ | ✔ | Score+Pair (ref=3) |

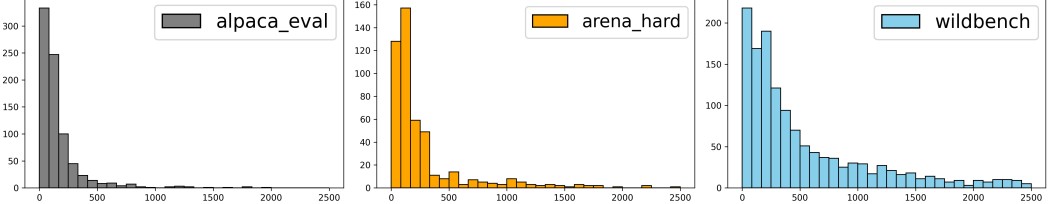

Figure 2: Distribution of query lengths in AlpacaEval, ArenaHard, and WildBench.

### 2.1 MINING CHALLENGING TASKS FROM WILDCHAT

We sourced tasks from the WildChat dataset (Zhao et al., 2024), which comprises one million human-chatbot conversations from real users. This dataset is particularly suited for conversion into an evaluation benchmark because it contains a diverse array of tasks that users expect LLMs to perform, such as writing assistance, coding, mathematics, data analysis, role playing, and planning.

**Basic filtering.** To control the quality and diversity of the selected tasks, we applied several filtering steps. First, we removed user queries that were either too short (less than 10 tokens) or excessively long (more than 3,000 tokens). We also excluded conversations with more than five user-chatbot turns to maintain focus and coherence in the tasks, as conversations exceeding five turns tend to contain multiple topics. Furthermore, we focused on English data and filtered out non-English tasks. Since our focus is more on evaluating the capabilities of LLMs rather than content moderation, we also removed toxic conversations. To ensure task diversity, we used sentence embeddings from SentenceBERT (Reimers & Gurevych, 2019) to calculate the cosine similarity between queries, discarding those with a high similarity score above 0.9. The threshold is determined by manual inspection. Lastly, to further enhance task diversity, we used a diverse user pool by retaining only the last conversation for each unique device, thus removing tasks from the same user that might require similar underlying skills.

**Difficulty annotation.** To identify challenging tasks that can distinguish the performance of different LLMs, we used GPT-4-Turbo (OpenAI, 2023), Claude-3-Sonnet, and Opus (Anthropic, 2024) to

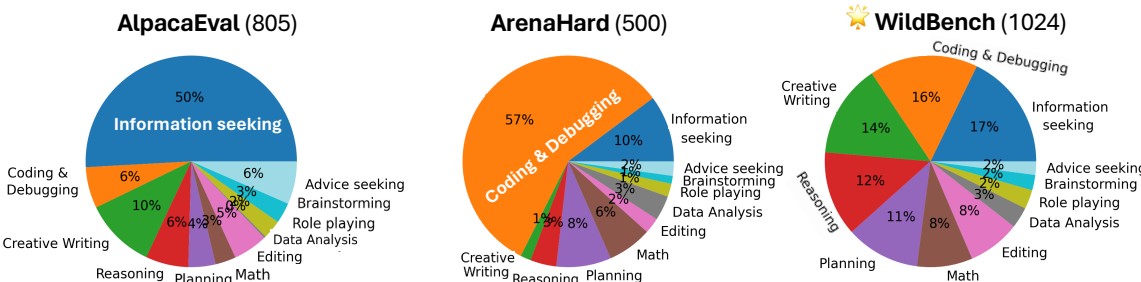

Figure 3: Distribution of task categories in AlpacaEval, ArenaHard, and WildBench.

analyze the required background knowledge and reasoning capabilities for each task. These models assigned a difficulty rating on a five-point scale (from "very easy" to "very hard"). Tasks rated as "very easy" or "easy" by all models were excluded. From the remaining pool, we randomly sampled 1,500 tasks to ensure that the distribution of task categories is similar to the original dataset.

**Human annotation.** To improve the quality of selected tasks, human annotation was used for quality control. We first used GPT-4-Turbo to summarize the intent of each query. These summaries were then used to help human reviewers remove nonsensical tasks. Finally, we retained 1,024 tasks for WILDBENCH. We also manually reviewed the tasks to ensure that they were challenging and diverse, covering a wide range of task categories. For the checklist questions, we verified that they were clear, interpretable, and relevant to the evaluation of LLM responses.

**Dynamic updates and data leakage prevention.** WILDBENCH is designed to be a dynamic benchmark that is updated regularly to reflect new types of user interactions. In fact, we have already released two versions of the benchmark (V1 in 2024 March and V2 in 2024 May), with similar curation process but on different iterations of WildChat data. To prevent potential data leakage for LLMs that use WildChat as part of their training or alignment, we coordinated with the WildChat team to ensure that the tasks we sample will not be publicly available in the WildChat dataset.

## 2.2 WILDBENCH STATISTICS

To better understand the composition of our evaluation, we analyze basic statistics and task categories.

**Basic statistics.** Table 1 compares the statistics of WILDBENCH to existing benchmarks AlpacaEval (Li et al., 2023b; Dubois et al., 2024), MT-Bench (Zheng et al., 2024), and ArenaHard (Li et al., 2024). Among these benchmarks, only ArenaHard and WILDBENCH are sourced from user queries in the wild ("RealUser"), rather than being curated by experts or through crowdsourcing. The difference between ArenaHard and our WildBench is that our data distribution aligns with real users' task categories, rather than overly focusing on coding and debugging as ArenaHard does.

**Long-context tasks.** WILDBENCH includes conversation histories of up to four turns per conversation, reflecting complex and extended user interactions that are facilitated by recent advancements in LLMs, with over 20% of conversations having more than two or more turns as shown in Figure 8. Additionally, as shown in Figure 2, WILDBENCH has longer query lengths, attributable to the extensive context provided by real user interactions captured in the dataset. This is because that GPT-4-Turbo, one of the chatbots behind WildChat, supports up to 128K context tokens and 4K output tokens. This capability exemplifies the importance of a dynamic, in-the-wild benchmark: as models evolve, they unlock new user applications. Thanks to these realistic user activities, WILDBENCH is a more suitable benchmark for testing the long-context problem solving abilities of LLMs.

**Task categories.** To enable a fine-grained analysis of LLM capabilities across varied tasks, we categorize the tasks into 12 categories based on previous analysis of ShareGPT queries (Ouyang et al., 2023) and our intent annotation of the tasks. Detailed descriptions about the 12 task categories are shown in Appendix A. The distribution of the task categories is shown in Figure 3. In this figure, we also compare to AlpacaEval and ArenaHard. Notably, WILDBENCH is more balanced compared to AlpacaEval and ArenaHard, which have over 50% of their tasks in Information seeking and Coding & Debugging categories, respectively.

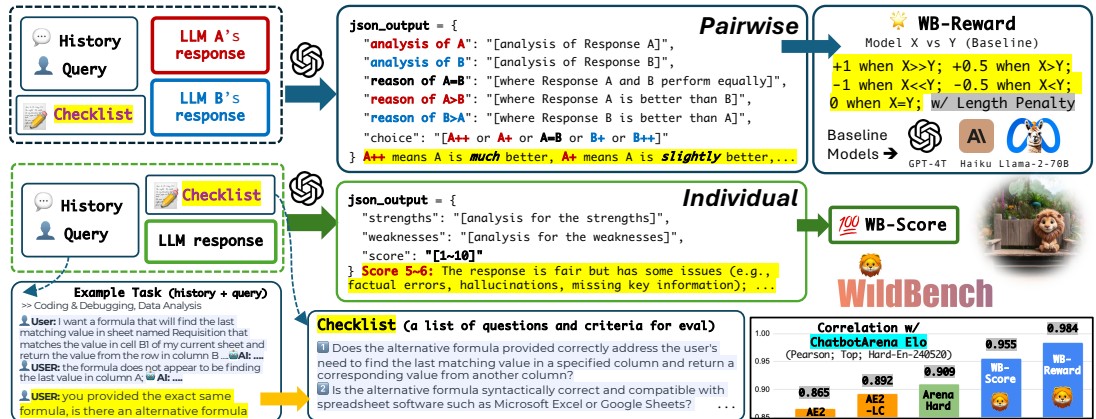

Figure 4: Evaluation framework for WILDBENCH. There are two metrics: WB-Score for individual evaluation and WB-Reward for pairwise evaluation. The checklist is used to guide the evaluation process. The length penalty is used to mitigate the length bias. WB-Reward and WB-Score both have strong correlations with human-based ranking of LLMs on Chatbot Arena.

## 3 AUTOMATIC EVALUATION WITH WILDBENCH

In this section, we introduce the evaluation process of LLMs using WILDBENCH. We first explain how we generate a checklist for each test query to enhance interpretability and reduce evaluation ambiguity in WILDBENCH. Then, we introduce two automatic metrics: WILDBENCH-Score and WILDBENCH-Reward. Finally, we discuss how we mitigate the length bias in the evaluation process.

### 3.1 INSTANCE-SPECIFIC CHECKLISTS

Powerful LLMs have been widely used as judges to evaluate the quality of LLM outputs in many automatic evaluation methods, such as AlpacaEval (Li et al., 2023b). However, even asking humans to judge which of the given two model outputs is better can be subjective and ambiguous. Moreover, such judgements provide limited information about the quality of the models. Without a constant, interpretable, and comprehensive evaluation standard, the results can be noisy and hard to interpret.

To address this issue, we generate a checklist for each test query in WILDBENCH to comprehensively evaluate the responses of different models. The checklist consists of 5-10 questions that are designed to be interpretable and easy to verify. We combine the responses of GPT-4-Turbo and Claude-3-Opus to finalize the checklists, thereby mitigating the bias of using a single LLM as the evaluator. These checklists have been manually reviewed and are used as part of the prompts for LLM judges to evaluate the responses of different models. An example of the checklist can be found in Figure 4. Taking the G20 example in Figure 1, here is a subset of checklist questions for the task:

---

**Example checklist for the G20 task example in Figure 1.**

- ✔ *Does the essay contain more than 1200 words as requested by the user?*
- ✔ *Is the language of the essay beautiful and poetic, incorporating extensive vocabulary as specified?*
- ✔ *Does the essay include a significant amount of factual and empirical data related to the impact of the G20 summit on the global economy, trade, and development?*
- ✔ *Are there references to the role of young people in shaping the future of the world within the context of the G20 summit?*
- ✔ *Does the essay include ancient Indian historical references as requested by the user?*
- ✔ *Is the essay structured in a clear and logical manner, facilitating an easy understanding of the discussed topics?*

---

### 3.2 PAIRWISE EVALUATION WITH WB-REWARD METRIC

WB-Reward is based on pairwise evaluation, which uses a GPT-4-Turbo judge to compare the responses of two LLMs to determine which one performs better on a given task, using a structured

checklist to guide the comparison. This metric provides straightforward comparisons among models and the intermediate outcomes of win/lose rates are easy to interpret.

**Step-by-step evaluation process.** In Figure 4, we detail the step-by-step evaluation process for pairwise comparison. First, we provide a chain of evaluation questions to guide the LLM judge to analyze the user query and the conversation history. The LLM then evaluates the two responses and also analyze where and why one is better than the other. Finally, we ask the LLM to make a final judgment on which response is better and why. This method is inspired by the evaluation process in human evaluation, where human judges are asked to provide detailed feedback on the quality of the responses before making a final decision. The full evaluation prompt can be found at Appendix D

**WB-Reward metric.** To compute the WB-Reward for a test model X against a baseline model Y, we assign rewards based on the comparison result: +1 if X is much better than Y, +0.5 if X is slightly better than Y, 0 for a tie, -0.5 for X is slightly worse than Y, and -1 for X is much worse than Y.

**Baseline LLMs for pairwise evaluation.** Using a single baseline model for pairwise evaluation can lead to noisy and biased evaluations. To mitigate this issue, we use three baseline models (GPT-4-Turbo-0429, Claude-3-Haiku, and Llama-2-70B-chat (Touvron et al., 2023)) to compute the rewards for each model. Our metric WB-Reward (Mix) is the average of the rewards from these three baselines on 1024 examples, providing a more robust performance evaluation on WILDBENCH.

**Mitigating length bias with a margin for ties.** Previous studies have shown that LLM judges tend to prefer longer responses (Dubois et al., 2024). To mitigate this bias, we propose a simple and intuitive length penalty method. If the winning response is longer than the losing one by a certain threshold ($K$ characters), we convert Slightly Win/Slightly Lose to a Tie. $K$ can be customized via our leaderboard web-page for personalized configuration. Setting $K = \infty$ will disable the length penalty. We designed this feature to support a more personalized and flexible leaderboard. For example, users who prefer shorter and more concise outputs can set a smaller K if they do not prioritize correlating perfectly with the general human-based model rankings on ChatbotArena. This choice allows for a customized leaderboard experience depending on user preferences.

## 3.3 INDIVIDUAL EVALUATION WITH WB-SCORE METRIC

Although pairwise evaluation provides a direct comparison between LLMs, it is usually more expensive and time-consuming than grading each individual LLM generation. To individually evaluate the performance of each model on WILDBENCH, we prompt GPT-4-Turbo to assign a score from 1 to 10 for each model's response. The full evaluation prompt can be found at Appendix E.

**Score definition.** To ensure a stable and consistent evaluation, we ask GPT-4-Turbo to evaluate the quality of each response based on the checklist and provide detailed strengths and weakness of each output before giving a score from 1 to 10. The scores are defined as follows:

- Score 1–2: The response is very poor and does not make sense at all.
- Score 3–4: The response is poor and does not help the user solve the problem meaningfully.
- Score 5–6: The response is fair but has issues (e.g., factual errors, hallucinations, missing key information).
- Score 7–8: The response is good but could be improved.
- Score 9–10: The response is perfect and provides helpful information to solve the problem.

**Score rescaling.** The WILDBENCH-Score is calculated as the average of the scores on all examples tested, where each score is first subtracted by 5 and then multiplied by 2 (i.e., $S' = (S - 5) \times 2$). A score of 5 represents a borderline acceptable response, so this rescaling can help to better differentiate the performance of models that can effectively solve the tasks.

## 4 RESULTS & ANALYSIS

We analyze the performance of different models on WILDBENCH. We first present the leaderboard analysis, then examine the length bias issue in the evaluation process, and finally discuss the correlation between WILDBENCH-Score and ChatbotArena Elo rating.

**Leaderboard features.** In Table 2, we present a subset of the results from our live leaderboard demo. For the most up-to-date results and more interactive features, such as customizing length penalties and viewing the detailed task-wise performance of each model, please refer to our live leaderboard. Our

Table 2: Evaluation results (subset) of LLMs using WILDBENCH and other benchmarks. Please refer to Figure 6-7 and demo website to view and interact with the full results.

| | Model names | WB-Reward (no length penalty) | | | | WB-Score | Arena Elo | Arena-Hard | AlpacaEval2 | |
|---|---|---|---|---|---|---|---|---|---|---|
| | | Mix | ◎GPT4T | ◎Haiku | ◎Llama2 | | | | LC | WR |
| 1 | GPT-4o-0513 🔒 | 35.7 | 1.5 | 46.3 | 59.3 | 65.3 | 1293 | - | 57.5 | 51.3 |
| 2 | ◎ GPT-4-Turbo-0409 🔒 | 34.6 | *0* | 45.3 | 58.4 | 64.7 | 1251 | 82.6 | 55.0 | 46.1 |
| 3 | GPT-4-Turbo-0125 🔒 | 29.9 | -4.4 | 38.8 | 55.2 | 63.3 | 1239 | 78.0 | - | - |
| 4 | Gemini-1.5-Pro 🔒 | 27.8 | -4.4 | 37.9 | 50 | 55.7 | - | - | - | - |
| 5 | Llama-3-70B-Inst | 21 | -19 | 31.9 | 50.2 | 60.4 | 1213 | 41.1 | 34.4 | 33.2 |
| 6 | Claude 3 Opus 🔒 | 20.1 | -20.4 | 34.3 | 46.3 | 63.1 | 1232 | 60.4 | 40.5 | 29.1 |
| 7 | Gemini-1.5-Flash 🔒 | 17.4 | -16.6 | 26.3 | 42.5 | 53.1 | - | - | - | - |
| 8 | Yi-1.5-34B-Chat | 16.8 | -18.3 | 24.1 | 44.5 | 57.8 | - | - | - | - |
| 10 | Llama3-Inst-8B-SimPO | 14 | -22.5 | 18.9 | 45.7 | 53.9 | - | 33.8 | 44.7 | 40.5 |
| 13 | Claude 3 Sonnet 🔒 | 7.2 | -31.6 | 19.4 | 33.9 | 55.5 | 1187 | 46.8 | 34.9 | 25.6 |
| 14 | Qwen1.5-72B-Chat | 4.4 | -34.8 | 13.1 | 34.7 | 56.5 | 1143 | 36.1 | 36.6 | 26.5 |
| 17 | Command-R-Plus 🔒 | 0.4 | -36.3 | 7.4 | 30.2 | 51.4 | 1155 | 33.1 | - | - |
| 20 | ◎ Claude 3 Haiku 🔒 | -8.5 | -46.9 | *0* | 21.4 | 50.4 | 1169 | 41.5 | - | - |
| 21 | Mistral-Large 🔒 | -10.5 | -48.1 | -4 | 20.5 | 54.2 | 1158 | 37.7 | 32.7 | 21.4 |
| 23 | StarlingLM-7B-beta | -11.9 | -48.7 | -5 | 18 | 46.8 | 1111 | 23.0 | - | - |
| 24 | Llama-3-8B-Inst | -14.6 | -49.8 | -9.7 | 15.7 | 45.7 | 1144 | 20.6 | 22.9 | 22.6 |
| 25 | Command-R 🔒 | -16 | -48.4 | -12.7 | 13.1 | 45.7 | 1106 | 17.0 | - | - |
| 26 | Mixtral-8x7B-Inst | -18.8 | -53.4 | -13.5 | 10.4 | 47.8 | 1114 | 23.4 | 23.7 | 18.3 |
| 27 | DBRX Inst | -21.6 | -57.3 | -16.3 | 8.7 | 48.9 | 1106 | 23.9 | 25.4 | 18.4 |
| 29 | Yi-1.5-6B-Chat | -24.3 | -55 | -19.9 | 2.1 | 39.6 | - | - | - | - |
| 30 | Mistral-7B-Inst-v0.2 | -25 | -58.1 | -22.4 | 5.5 | 43.4 | 1071 | - | 17.1 | 14.7 |
| 32 | Tulu-2-dpo-70b | -25.4 | -59.3 | -20.3 | 3.3 | 45.2 | 1099 | 15.0 | 21.2 | 16.0 |
| 33 | ◎ Llama-2-70B-chat | -26.8 | -56.9 | -23.6 | *0* | 39.2 | 1070 | 11.6 | 14.7 | 13.9 |
| 34 | Qwen1.5-7B-Chat | -27 | -57.7 | -23 | -0.2 | 40 | 1059 | - | 14.7 | 11.8 |
| 35 | Phi-3-medium-128k | -33.3 | -66.4 | -30 | -3.6 | 42.1 | - | - | - | - |
| 36 | GPT-3.5-turbo-0125 | -33.5 | -66.3 | -30 | -4.1 | 42.1 | 1105 | 23.3 | - | - |
| 38 | Llama-2-7B-chat | -48 | -71.8 | -44.6 | -27.8 | 27.6 | 1012 | 4.6 | 5.4 | 5.0 |
| 39 | Gemma-7B-it | -57 | -78.4 | -55.8 | -36.8 | 23.9 | 1047 | 7.5 | 10.4 | 6.9 |
| 40 | Gemma-2B-it | -74.1 | -87.8 | -73.6 | -60.8 | 6.2 | 980 | 3.0 | 5.4 | 3.4 |

live leaderboard also supports exploring data and comparing model outputs side by side to understand the strengths and weaknesses of each model.

By using three baseline models of varying performance levels (GPT-4-Turbo > Claude 3 Haiku > Llama-2-70B-chat), we observe that the tested models can be naturally grouped into three tiers based on their performance. Tier 1 models outperform Claude 3 Haiku, Tier 2 models outperform Llama-2-70B-chat but are worse than Claude 3 Haiku, and Tier 3 models are worse than Llama-2-70B-chat.

## 4.1 LEADERBOARD ANALYSIS

**Where are the gaps between models?** A unique feature of the WILDBENCH leaderboard is the ability to compare models across different task categories, which enables us to identify the strengths and weaknesses of each model on different types of tasks. In Figure 5, we select a set of popular models for analysis: Llama-3-8B-Inst (Meta, 2023), Llama-3-8B-Inst-SimPO (Meng et al., 2024b), Yi-1.5-34B-chat (AI et al., 2024), Llama-3-70B-Inst, GPT-4-Turbo-0409, and Claude 3 Opus. We show their performance in WB-Score across five task categories (merged from the 12 categories shown in Figure 3). Larger models like GPT-4-Turbo-0409 and Claude 3 Opus perform well across all task categories, while open LLMs like Llama-3-8B-Inst and Yi-1.5-34B-chat show weaker performance on coding and math-related tasks.

**Will an 8B model outperform a 70B model?** On the AlpacaEval-2.0 leaderboard, Llama-3-8B-Inst-SimPO (LC=44.7%) significantly outperforms Llama-3-70B-Inst (LC=34.4%) (Meng et al., 2024a), which is surprising and differs from our results. As shown in both Table 2 and Figure 5, our results indicate that Llama-3-8B-Inst-SimPO is generally still worse than Yi-34B-chat and Llama-3-70B-Inst. However, on information-seeking and creative tasks, Llama-3-8B-Inst-SimPO performs comparably to Llama-3-70B-Inst. Thus, we believe AlpacaEval's evaluation results underestimate the performance of Llama-3-70B-Inst due to task selection bias in addition to the weakness of their evaluation prompting method. While the performance of Llama-3-8B-Inst-SimPO is not as good as it

Table 3: Correlation with Chatbot ArenaElo Elo (Hard-En-240520) of alignment benchmarks.

| Metric | P-Cor$_{top}$ | P-Cor$_{all}$ | S-Cor$_{all}$ | K-Cor$_{all}$ | Metric | P-Cor$_{top}$ | P-Cor$_{all}$ | S-Cor$_{all}$ |
|---|---|---|---|---|---|---|---|---|
| ArenaElo (Hard-En) | 1.000 | 1.000 | 1.000 | 1.000 | Avg Length | 0.472 | 0.554 | 0.376 |
| Arena-Hard | *0.909* | 0.925 | *0.965* | *0.890* | WB-Reward$^{llama}_{\infty}$ | 0.976 | 0.965 | 0.965 |
| AlpacaEval2-LC | 0.892 | 0.951 | 0.924 | 0.818 | WB-Reward$^{gpt4t}_{\infty}$ | 0.974 | 0.961 | 0.965 |
| AlpacaEval2 | 0.865 | *0.952* | 0.960 | 0.868 | WB-Reward$^{haiku}_{\infty}$ | **0.985** | **0.974** | **0.982** |
| WB-Score | 0.955 | 0.940 | 0.943 | 0.846 | WB-Reward$^{llama}_{500}$ | 0.977 | 0.969 | 0.961 |
| WB-Reward$^{mix}_{\infty}$ | **0.984** | 0.973 | **0.978** | **0.912** | WB-Reward$^{gpt4t}_{500}$ | **0.992** | 0.973 | 0.969 |
| WB-Reward$^{mix}_{500}$ | **0.984** | **0.976** | 0.974 | **0.912** | WB-Reward$^{haiku}_{500}$ | 0.973 | **0.976** | **0.974** |

seems on AlpacaEval-2.0, it is indeed the best 8B model in our evaluation and outperforms some other larger models. Interestingly, Llama-3-8B-Inst-SimPO consistently improves the performance of Llama-3-8B-Inst on all task categories, resulting in a similar shape on the radar plot in Figure 5.

**Are longer responses always better?** WILD-BENCH is robust to length bias. For example, Llama-2-70B-chat and Llama-3-70B-Inst have similar output lengths (2,965 vs 2,983 chars), yet Llama-3-70B-Inst ranks 5th while Llama-2-70B-chat ranks 33rd on the leaderboard of 40 models. Additionally, Yi-1.5-6B's output length is the 4th longest among the 40 models (3,322 characters), but it ranks 29th on the leaderboard. This suggests that the WILDBENCH evaluation is not biased towards longer responses, with response quality being the most important factor in the evaluation process. Additionally, we use a length penalty to ensure that longer responses are not always favored, and users can customize the length penalty to adjust the trade-off between response length and quality according to their needs. This feature is available on our live leaderboard and is illustrated in Figure 6.

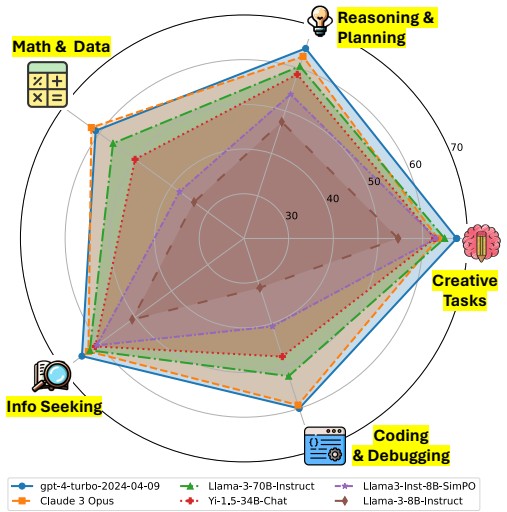

Figure 5: Performance breakdown by task category of 6 models on WILDBENCH.

## 4.2 CORRELATION TO HUMAN JUDGMENT

To analyze how well WILDBENCH evaluation correlates with human judgment, we compare our results to the ChatbotArena Elo rating generated by large-scale online human evaluations. Focusing on hard prompts, we use the Elo ratings from the Hard-English version released on May 20, 2024.

We compare our WB-Reward and WB-Score with three other metrics: AlpacaEval winrate (WR), length-controlled winrate (LC), and ArenaHard scores. We use three correlation metrics: Pearson correlation (P-Cor), Spearman correlation (S-Cor), and Kendall's tau correlation (K-Cor). To ensure a fair comparison, we consider all models that have all four metrics available in Table 2, which results in 14 models. To distinguish the top-performing models, we also consider the top 6 models, denoting their correlation metrics as P-Cor$_{top}$, and P-Cor$_{all}$ respectively. The reason why we care about the correlation on top-ranking models is that models released in the future are likely to compete with the top models, so the Pearson correlation in this range is more important from the perspective of predicting the future application of a metric. The analysis results are shown in Table 3.

Both WB-Reward and WB-Score show strong correlations with the human-based Elo rating, particularly for the top-performing models, achieving the best correlation among all other automatic metrics. Among using different baseline models for pairwise evaluation, we find that using Haiku as the baseline model yields the best correlation. These results suggest that the WILDBENCH evaluation correlates well with human judgment in ranking model performance as an automatic metric.

### 4.3 ABLATION STUDIES AND DISCUSSIONS.

**Checklists.** In our ablation study on the impact of checklists, we compared model performance with and without checklists by removing the associated parts from the prompt templates. The results indicate that incorporating checklists improves the final correlation with human preferences. Specifically, the WB-Score without checklists achieves a Pearson correlation of 0.905 (for all models), which is lower than the 0.925 correlation achieved when using checklists.

**Length penalties.** We experimented with different $K$ (100, 200, 500, 1000, $\inf$) in the length penalty method. We found that $K = 500$ is the best choice, as it achieves the highest correlation with human judgments. This result suggests that the length penalty method is effective in mitigating the length bias in LLM evaluations.

**Do multiple LLMs as judges help?** How much do multiple LLMs help? We experimented with using GPT-4, Claude 3 Opus, and Mistral-Large as LLM judges. Our experiments revealed that these LLM judges produced very similar results, thereby exerting minimal influence on the final relative ranking of LLMs. Considering to reduce the cost of evaluation and faster turnaround time, we recommend using a single LLM as a judge in practice. In the future versions, we will explore more efficient ways to use multiple LLMs as judges, for example, by using different judge LLMs for different tasks that are best suited to their strengths.

**Data distribution.** How do we explain that WildBench has a different distribution compared to ChatbotArena's platform but still shows a strong correlation, even better than ArenaHard? The objective of WildBench is to evaluate LLMs on challenging tasks from real users. The ArenaElo we use for comparison is derived from the hard-English split in ChatbotArena, where human users submit tasks and vote. Thus, both WildBench and ChatbotArena aim to address the same goal. While it is practically impossible to match the exact distribution of users and tasks between the two—given that WildChat users are anonymous and ChatbotArena does not publicize its data—both are sourced from real users on the web. Consequently, this represents the best possible approach for correlating our LLM ratings with human-based ratings.

**Two complementary metrics: WB-Reward & WB-Score**. Both metrics use checklists and a CoT-style prompt for evaluation, utilizing the same testing data. The key differences are in their methodologies: **WB-Score:** Evaluates each model's outputs individually on a scale of 1-10, with detailed explanations for each score (see Appendix); **WB-Reward:** Compares a model's outputs to those of three baseline models at different performance levels for a comprehensive evaluation. Pairwise evaluations can be coarse, but using three baseline models and refined pairwise choices (e.g., much better or slightly better) mitigates this. WB-Score provides a universal score comparable across models using the same evaluation templates and checklists. Additionally, WB-Score is cheaper and faster to run (10 minutes, $5) compared to WB-Reward, which requires 3-4 times the cost due to multiple baselines. Both metrics have their strengths and weaknesses. We use both to build our official leaderboard, allowing users to choose the most suitable metrics for their experiments.

## 5 RELATED WORKS

**Close-ended benchmarks.** Close-ended benchmarks typically consist of multiple-choice questions and have been widely used to evaluate LLMs authors (2022). For example, MMLU (Hendrycks et al., 2020) includes multi-choice questions across various subject areas. Its variants include CMMLU (Li et al., 2023a) for Chinese, KMMLU (Son et al., 2024) for Korean, and MMLU-Pro (Wang et al., 2024) for more challenging evaluation. GPQA (Rein et al., 2023) is another close-ended benchmark designed to be challenging even for humans with internet access. Specialized benchmarks with ground-truth answers, such as GSM8K (Cobbe et al., 2021) and MATH (Hendrycks et al., 2021), also fall into this category. While these benchmarks focus on close-form answers, our work evaluates LLMs' ability to generate free-form responses and engage in conversations with users.

**Expert-curated and crowdsourced data.** Several open-ended generation benchmarks rely on data curated by human experts or crowdsourcing workers. For instance, MT-Bench (Zheng et al., 2024) manually creates examples for predefined categories. AlpacaEval (Li et al., 2023b) is based on author-written examples (Dubois et al., 2023; Taori et al., 2023; Wang et al., 2022), which primarily consists of simple instructions such as rewriting tasks.

**In-the-wild data.** A key feature of our work is that its underlying data is sourced from real-world use cases, ensuring alignment with actual LLM use cases. Notable benchmarks using real-world data include ChatbotArena (Zheng et al., 2024; Chiang et al., 2024), where users input their questions and choose the better response from two LLMs. However, ChatbotArena relies on extensive human feedback. WildVision (Lu et al., 2024) is a similar project but designed for vision language models. ArenaHard (Li et al., 2024) is another work that selects user queries from ChatbotArena to construct a benchmark for automatic evaluation.

**Evaluation methods.** Evaluating open-ended generation poses challenges due to the lack of a single valid ground truth. Human evaluation, though reliable, is expensive and time-consuming. To reduce costs and enable fast evaluation, powerful LLMs are often used as judges, as seen in benchmarks like MT-Bench, AlpacaEval, ArenaHard, and our own. Evaluation methods include single-system grading, which assigns scores to individual outputs, and pairwise comparisons, which compare outputs of two systems to compute win rates. Pairwise comparisons, while more expensive, can highlight subtle differences across systems (Zheng et al., 2024). To mitigate self-selection bias where an LLM prefers its own outputs (Panickssery et al., 2024), we use checklists generated from multiple LLMs, similar to InfoBench (Qin et al., 2024). In addition, we ask LLM judges generate structured explanations that enable human verification for further calibration, inspired by Just-Eval (Lin et al., 2023). There are also local evaluators that can be used to evaluate LLMs with our WILDBENCH with open-weight LLMs, such as TIGERScore (Jiang et al., 2023) and Prometheus (Kim et al., 2024).

**Data leakage prevention.** Publicly available benchmarks risk contamination from LLMs trained on such data. GPQA includes a special string to help LLM developers filter out its data (Rein et al., 2023), yet indirect leakage through cited examples remains possible. To mitigate this, we reserve a subset of WildChat that is never released publicly, which keeps its expert-curated evaluation data private. However, WILDBENCH provides a public validation set and details the benchmark construction process for greater transparency.

**Other dimensions for evaluation.** While our focus is on evaluating LLM capabilities, other evaluation dimensions, such as safety (Mazeika et al., 2024; Jiang et al., 2024), fairness (Gallegos et al., 2024), logical reasoning (Lin et al., 2024), agentic planning (Liu et al., 2023; Mialon et al., 2023; Lin et al., 2022), and hallucination detection (Min et al., 2023; Mishra et al., 2024; Hong et al., 2024), are equally important.

## 6    CONCLUSION AND FUTURE DIRECTIONS

In this work, we introduced WILDBENCH, a benchmark designed to evaluate LLMs using real-world user queries. An important feature of WILDBENCH data is the nature of in-the-wild user queries with natural task distribution. To evaluate LLM performance using the collected data, we introduced a CoT-like LLM-as-judge method to improve the interpretability of evaluations and reduce ambiguity. We also incorporated a length penalty method to mitigate the length bias in LLM-as-judge evaluations. Experiments show that our primary metrics, WB-Reward and WB-Score, have very strong correlations with human judgments, surpassing existing evaluations.

We present extensive experiments and analyses, showcasing the performance of a wide range of 40 LLMs, including both proprietary and public ones, on the WILDBENCH benchmark. By providing a detailed breakdown of scores across different task categories, WILDBENCH offers insights on the strengths and weaknesses of different models. By introducing WILDBENCH, we aim to provide a realistic, dynamic, and contamination-resilient evaluation framework that accurately reflects the capabilities of LLMs. We will actively maintain the project for continually evaluating new LLMs with unseen tasks over time.

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

**Appendix**

## A  TASK CATEGORIES

In Section 2.2 we mentioned that tasks are categorized into 12 categories to enable fine-grained analysis of LLM capabilities. The definition of these task categories are as follows.

- **Information seeking** - Users ask for specific information or facts about various topics.
- **Reasoning** - Queries require logical thinking, problem-solving, or processing of complex ideas.
- **Planning** - Users need assistance in creating plans or strategies for activities and projects.
- **Editing** - Involves editing, rephrasing, proofreading, or other tasks related to the composition of general written content.
- **Coding & Debugging** - Users seek help with writing, reviewing, or fixing code in programming.
- **Math** - Queries related to mathematical concepts, problems, and calculations.
- **Role playing** - Users engage in scenarios requiring ChatGPT to adopt a character or persona.
- **Data Analysis** - Requests involve interpreting data, statistics, or performing analytical tasks.
- **Creative Writing** - Users seek assistance with crafting stories, poems, or other creative texts.
- **Advice seeking** - Users ask for recommendations or guidance on various personal or professional issues.
- **Brainstorming** - Involves generating ideas, creative thinking, or exploring possibilities.
- **Others** - Any queries that do not fit into the above categories or are of a miscellaneous nature.

We consolidate the original categories into five major groups for easier task-wise analysis. Specifically, we combine "Information seeking" and "Advice seeking" into "Info Seeking"; "Math" and "Data Analysis" into "Math & Data"; and "Reasoning" and "Planning" into "Reasoning & Planning." The remaining types are grouped under "Creative Tasks." These consolidated groups are illustrated in Figure 5.

Please note that the following links are allenai for double-blind review, which we will update after the review process. The supplementary zip file contains the source code for the evaluation scripts, the leaderboard, and the data.

Figure 8: Distribution of the number of turns in WildBench.

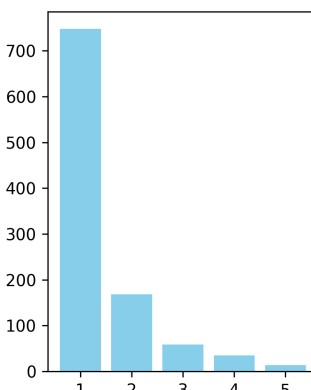

## B  MORE INFORMATION ON WILDBENCH DATA

The distribution of the number of turns in WILDBENCH can be found in Figure 8. The dataset documentation, metadata, and the public subset of WILDBENCH can be found at https://huggingface.co/datasets/allenai/WildBench/viewer/v2. We release the data under AI2's ImpACT license as a low-risk artifact, and we bear all responsibility in case of rights violations. We will ensure that the dataset will be available for a long time and maintain the data by continuously updating it.

## C  MORE INFORMATION ON WILDBENCH EVALUATION

Our evaluation results on the public subset of WILDBENCH can be reproduced using evaluation scripts available at https://github.com/allenai/WildBench/. We have included generation script for each model under the folder https://github.com/allenai/WildBench/tree/main/scripts, and the scripts for evaluating generations can be found at https://github.com/allenai/WildBench/tree/main/evaluation.

## D  PROMPT TEMPLATE FOR PAIRWISE EVALUATION METRIC WB-REWARD

The prompt template for pairwise evaluation is shown below. It can be divided into three sections: the first section provides the high-level instruction, the task to be tested, and two model outputs; the

second section specifies the checklist and the rules; and the last section instructs the LLM judge to follow the step-by-step evaluation process as detailed in Section 3.2

```
# Instruction
You are an expert evaluator. Your task is to evaluate the quality of
↪  the responses generated by two AI models. We will provide you with
↪  the user query and a pair of AI-generated responses (Response A and
↪  B). You should first read the user query and the conversation
↪  history carefully for analyzing the task, and then evaluate the
↪  quality of the responses based on and rules provided below.

# Conversation between User and AI

## History
<|begin_of_history|>
{$history}
<|end_of_history|>

## Current User Query
<|begin_of_query|>
{$user_query}
<|end_of_query|>

## Response A
<|begin_of_response_A|>
{$candidate_A}
<|end_of_response_A|>

## Response B
<|begin_of_response_B|>
{$candidate_B}
<|end_of_response_B|>
```

```
# Evaluation

## Checklist

<|begin_of_checklist|>
{$checklist}
<|end_of_checklist|>

Please use this checklist to guide your evaluation, but do not limit
↪  your assessment to the checklist.

## Rules

You should compare the above two responses based on your analysis of
↪  the user queries and the conversation history. You should first
↪  write down your analysis and the checklist that you used for the
↪  evaluation, and then provide your assessment according to the
↪  checklist. There are five choices to give your final assessment:
↪  ["A++", "A+", "A=B", "B+", "B++"], which correspond to the
↪  following meanings:

- `A++`: Response A is much better than Response B.
- `A+`: Response A is only slightly better than Response B.
- `A=B`: Response A and B are of the same quality. Please use this
↪  choice sparingly.
- `B+`: Response B is only slightly better than Response A.
- `B++`: Response B is much better than Response A.
```

```
## Output Format
First, please output your analysis for each model response, and then
↪   summarize your assessment to three aspects: "reason A=B", "reason
↪   A>B", and "reason B>A", and finally make your choice for the final
↪   assessment.

Please provide your evaluation results in the following json format by
↪   filling in the placeholders in []:
```
{
    "analysis of A": "[analysis of Response A]",
    "analysis of B": "[analysis of Response B]",
    "reason of A=B": "[where Response A and B perform equally well]",
    "reason of A>B": "[where Response A is better than Response B]",
    "reason of B>A": "[where Response B is better than Response A]",
    "choice": "[A++ or A+ or A=B or B+ or B++]",
}
```
```

# E   PROMPT TEMPLATE FOR INDIVIDUAL EVALUATION METRIC WB-SCORE

The prompt template for individual evaluation is shown below. It can be similarly divided into three
sections: the first section provides the high-level instruction, the task to be tested, and the model
output; the second section specifies the checklist and the rules; and the last section instructs the LLM
judge to follow the step-by-step evaluation process as detailed in Section 3.3.

```
# Instruction

You are an expert evaluator. Your task is to evaluate the quality of
↪   the responses generated by AI models.
We will provide you with the user query and an AI-generated responses.
You should first read the user query and the conversation history
↪   carefully for analyzing the task, and then evaluate the quality of
↪   the responses based on and rules provided below.

# Conversation between User and AI

## History
<|begin_of_history|>

{$history}

<|end_of_history|>

## Current User Query
<|begin_of_query|>

{$user_query}

<|end_of_query|>

## AI Response
<|begin_of_response|>

{$model_output}

<|end_of_response|>
```

```
# Evaluation

## Checklist

<|begin_of_checklist|>

{$checklist}

<|end_of_checklist|>

Please use this checklist to guide your evaluation, but do not limit
↪  your assessment to the checklist.

## Rules

You should compare the above response based on your analysis of the
↪  user queries and the conversation history.
You should first write down your analysis and the checklist that you
↪  used for the evaluation, and then provide your assessment according
↪  to the checklist.
The scores are in the range of 1~10, where 1 means the response is very
↪  poor and 10 means the response is perfect.
Here are more detailed criteria for the scores:

- Score 1~2: The response is very poor and does not make sense at all.
- Score 3~4: The response is poor and does help user solve the problem
↪   in a meaningful way.
- Score 5~6: The response is fair but has some issues (e.g., factual
↪   errors, hallucinations, missing key information).
- Score 7~8: The response is good enough but could be improved in some
↪   ways.
- Score 9~10: The response is perfect and provides helpful information
↪   that can help user solve the problem.
```

```
## Output Format
First, please output your analysis for the model response, and then
↪  summarize your assessment to two aspects: "strengths" and
↪  "weaknesses"; Finally, please write down your rating for the
↪  assessment.

Please provide your evaluation results in the following json format by
↪  filling in the placeholders in []:
```
{
    "strengths": "[analysis for the strengths of the response]",
    "weaknesses": "[analysis for the weaknesses of the response]",
    "score": "[1~10]"
}
```
```

## F  FULL WILDBENCH LEADERBOARD

The full WILDBENCH leaderboard as of Jun 5, 2024 can be found in Figure 6; The updated leaderboard as of Sept 1, 2024 can be found in Figure 7. Note that we used a new metric named WB-Elo that is based on merging WB-Reward and WB-Score to a collection of pairwise comparisons and perform Elo rating updates on top of existing LMSYS Elo rating, thus we can have a faster and more stable leaderboard update. You can view and interact with the latest results on our leaderboard on our website at https://huggingface.co/spaces/allenai/WildBench

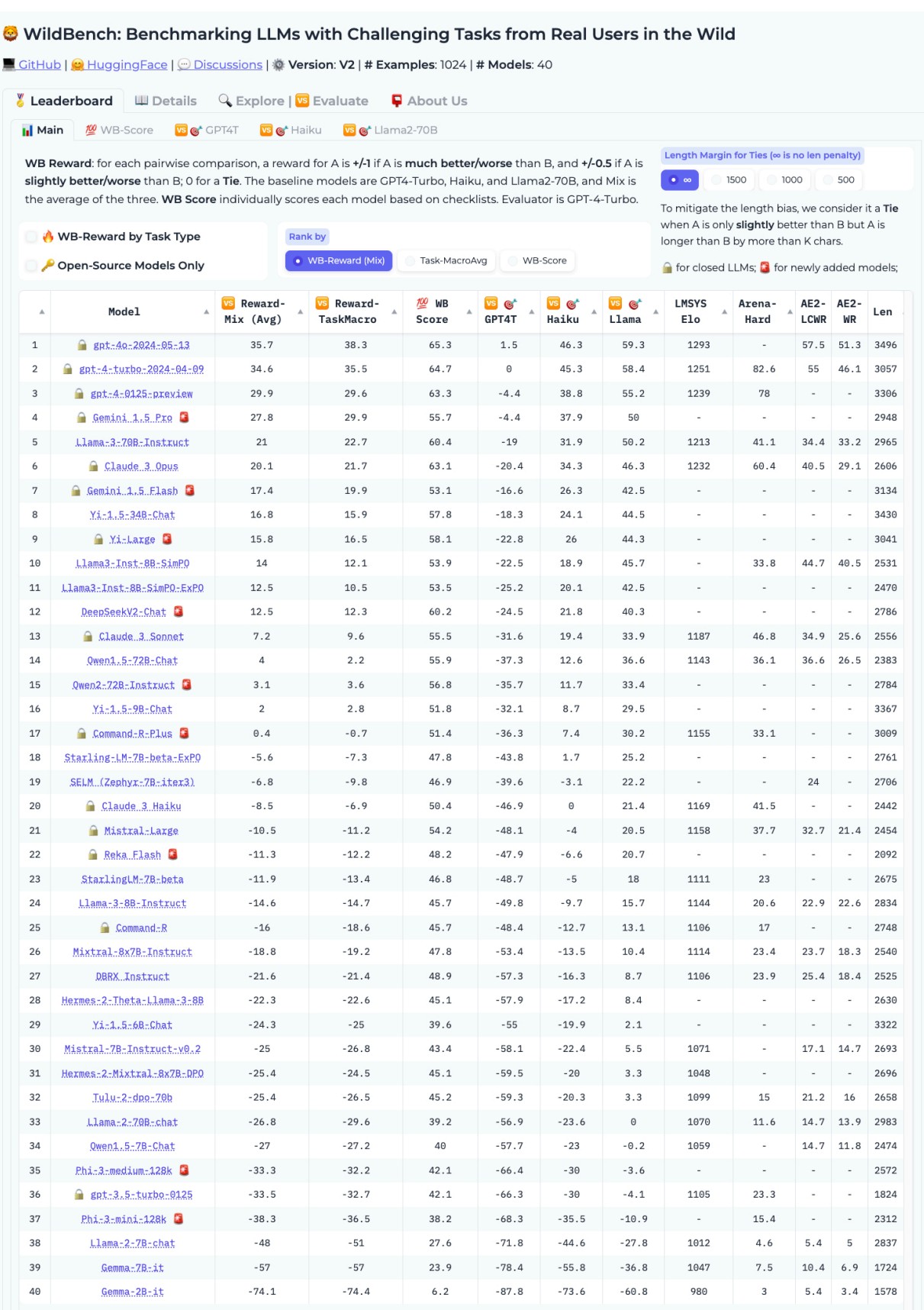

Figure 6: Leaderboard of WildBench (2024 Jun 5th)

| | Model | 🏆 WB-Elo (LC) | ℹ️ Info Seek | 📝 Creative | 🖥️ Code & Debug | 📊 Math & Data | 🔄 Reason & Plan | 💯 Score | 🏆 WB-Elo (Raw) | Len |
|---|---|---|---|---|---|---|---|---|---|---|
| 1 | 🔒 gpt-4o-2024-05-13 | 1227.1 | 58.6 | 59.1 | 60.5 | 57.3 | 60.2 | 59.3 | 1236.7 | 3723 |
| 2 | 🔒 Claude-3.5-Sonnet 🔴 | 1215.4 | 55.5 | 55.6 | 56.5 | 50.2 | 55.6 | 54.7 | 1221.9 | 2911 |
| 3 | 🔒 Gemini-1.5-Pro | 1214.6 | 52.2 | 55.1 | 55.2 | 48.6 | 53.7 | 53 | 1220.3 | 3247 |
| 4 | 🔒 gpt-4-turbo-2024-04-09 | 1209.6 | 57.2 | 58.7 | 55.1 | 51 | 56.2 | 55.2 | 1217.1 | 3093 |
| 5 | 🔒 Yi-Large-Preview | 1208.9 | 57.7 | 57.6 | 54.3 | 51.9 | 56.6 | 55.3 | 1214.1 | 3512 |
| 6 | 🔑 DeepSeek-V2-Chat (0628 API) | 1199.1 | 52.7 | 56.4 | 55 | 51.4 | 54.8 | 54 | 1207.2 | 3252 |
| 7 | 🔒 gpt-4-0125-preview | 1197.3 | 54.4 | 57.6 | 52.9 | 45.8 | 53.5 | 52.3 | 1205.9 | 3335 |
| 8 | 🔒 Claude-3-Opus | 1196.3 | 53.5 | 53 | 53.3 | 46.7 | 52.5 | 51.7 | 1202.5 | 2685 |
| 9 | 🔒 Gemini-1.5-Flash | 1192 | 48.7 | 51.7 | 48.7 | 45.3 | 50.8 | 48.9 | 1196.8 | 3654 |
| 10 | 🔑 Llama-3-70B-Instruct | 1187.5 | 52.3 | 54.3 | 44.7 | 42.1 | 50.1 | 47.8 | 1193.6 | 3046 |
| 11 | 🔑 DeepSeek-V2-Coder (0614 API) | 1184.9 | 40 | 40.8 | 48.9 | 46.4 | 47.2 | 45.7 | 1175.9 | 2580 |
| 12 | 🔒 Yi-Large | 1181.8 | 51 | 51.8 | 47.7 | 44.5 | 51.3 | 48.9 | 1186.4 | 3095 |
| 13 | 🔑 Athene-70B 🔴 | 1180.7 | 60.8 | 60.4 | 59 | 57.1 | 61 | 59.5 | 1198.3 | 3175 |
| 14 | 🔑 Nemotron-4-340B-Inst | 1178.6 | 53 | 53.3 | 46.3 | 40.8 | 49.1 | 47.7 | 1182.3 | 2754 |
| 15 | 🔑 Gemma-2-27B-it 🔴 | 1176.4 | 50.5 | 53.6 | 47 | 43.9 | 50.6 | 48.5 | 1181 | 2924 |
| 16 | 🔒 Mistral-Large-2 🔴 | 1176.3 | 57.4 | 58.9 | 53.8 | 52.7 | 57.2 | 55.6 | 1190.5 | 3503 |
| 17 | 🔒 Claude-3-Sonnet | 1174.7 | 47.1 | 46.3 | 46.1 | 40.6 | 47.4 | 45.5 | 1176.4 | 2670 |
| 18 | 🔒 gpt-4o-mini-2024-07-18 🔴 | 1173.5 | 57.4 | 60.1 | 57.2 | 54 | 58.2 | 57.1 | 1193.2 | 3648 |
| 19 | 🔑 Qwen2-72B-Instruct | 1172.3 | 49.5 | 49.9 | 39.8 | 41 | 46.8 | 44.5 | 1176.7 | 2856 |
| 20 | 🔒 Reka-Core | 1170.4 | 52.3 | 55.5 | 40.6 | 40.3 | 48 | 45.9 | 1174.1 | 2592 |
| 21 | 🔑 gemma-2-9b-it-SimPO 🔴 | 1166.6 | 56.5 | 58 | 50.9 | 48.6 | 55.6 | 53.3 | 1186.5 | 4277 |
| 22 | 🔑 gemma-2-9b-it-DPO 🔴 | 1166.6 | 58.2 | 59.1 | 50.5 | 47.1 | 55.5 | 53.2 | 1184.4 | 3982 |
| 23 | 🔑 Yi-1.5-34B-Chat | 1159.6 | 50.3 | 53.5 | 42.1 | 39.4 | 48.1 | 45.6 | 1164.4 | 3523 |
| 24 | 🔒 Claude-3-Haiku | 1159.1 | 45.3 | 42.9 | 37 | 31.4 | 41.3 | 38.9 | 1159.3 | 2601 |
| 25 | 🔑 Mistral-Nemo-Inst (12B) 🔴 | 1158.6 | 51.9 | 54.6 | 39.7 | 35.6 | 47.4 | 44.4 | 1166.9 | 3318 |
| 26 | 🔒 Mistral-Large | 1157 | 46.1 | 49.7 | 33.7 | 30.9 | 41.8 | 38.9 | 1159.5 | 2514 |
| 27 | 🔑 Gemma-2-9B-it 🔴 | 1156.4 | 49 | 51 | 36.7 | 36.4 | 46.7 | 42.7 | 1159.9 | 2802 |
| 28 | 🔒 Command-R-Plus | 1151.4 | 49.2 | 52.6 | 28.4 | 23.5 | 41.9 | 36.8 | 1153.3 | 3293 |
| 29 | 🔑 GLM-4-9B-Chat | 1148.5 | 46.3 | 47.8 | 35.4 | 29.8 | 42.5 | 39.1 | 1153.9 | 3692 |
| 30 | 🔑 Magpie-8B-Align-v0.1 🔴 | 1148.4 | 48.9 | 49.2 | 33.7 | 29.8 | 42.7 | 39.3 | 1154.8 | 3107 |
| 31 | 🔑 Yi-1.5-9B-Chat | 1148 | 42.6 | 45.6 | 35 | 32.2 | 42.4 | 38.7 | 1154.2 | 3468 |
| 32 | 🔑 Llama3-Inst-8B-SimPO | 1147.5 | 47.9 | 50.6 | 31.8 | 24 | 40.9 | 37 | 1152.2 | 2541 |
| 33 | 🔑 Llama3-Inst-8B-SimPO-v0.2 | 1147.4 | 47.9 | 51.8 | 31.5 | 24.4 | 40.7 | 37.2 | 1151.3 | 2533 |
| 34 | 🔑 Qwen1.5-72B-Chat | 1147.4 | 48.2 | 50.4 | 35.4 | 29.8 | 43.5 | 39.9 | 1150 | 2392 |
| 35 | 🔑 Llama3-Inst-8B-SimPO-ExPO | 1145.5 | 47.3 | 49.1 | 28.6 | 21.2 | 39.5 | 35 | 1147.3 | 2480 |
| 36 | 🔑 SELM (Llama3-8B-Inst-iter3) | 1144 | 46.1 | 51.1 | 27.3 | 23.5 | 39.8 | 35.3 | 1148.9 | 2913 |
| 37 | 🔑 Phi-3-medium-128k | 1139.5 | 35.7 | 33.2 | 18.2 | 23 | 32.3 | 27.3 | 1128.2 | 2849 |
| 38 | 🔑 Llama-3-8B-Instruct | 1139.5 | 39.3 | 43.6 | 22 | 17 | 34.4 | 29.2 | 1138.6 | 2975 |
| 39 | 🔑 Hermes-2-Theta-Llama-3-8B | 1137.4 | 41.6 | 39.8 | 23.1 | 18.7 | 33.7 | 29.6 | 1137.7 | 2742 |
| 40 | 🔑 Starling-LM-7B-beta-ExPO | 1136 | 42.9 | 44.3 | 25.3 | 18.6 | 36.3 | 31.6 | 1137.8 | 2835 |
| 41 | 🔑 SELM (Zephyr-7B-iter3) | 1134.3 | 41 | 44.7 | 11 | 12.7 | 31.6 | 25.1 | 1126.5 | 2823 |
| 42 | 🔒 Reka-Flash | 1132.7 | 41.5 | 42.4 | 22.1 | 20.5 | 35 | 30.4 | 1132.1 | 2103 |
| 43 | 🔑 Gemma-2-2B-it 🔴 | 1129.7 | 39.9 | 43.6 | 17.9 | 15.8 | 33.8 | 27.8 | 1128.8 | 3589 |
| 44 | 🔒 gpt-3.5-turbo-0125 | 1129.2 | 36.5 | 37.4 | 26.5 | 21.6 | 33.4 | 30 | 1122.7 | 1844 |
| 45 | 🔑 DBRX-Instruct | 1128.5 | 41.1 | 42.3 | 26.4 | 24.5 | 36.2 | 32.6 | 1129.4 | 2576 |
| 46 | 💎🔑 Neo-7B-Instruct-ExPO | 1126.6 | 34.9 | 38.5 | 12.8 | 12.6 | 28.7 | 23.1 | 1116 | 4107 |
| 47 | 💎🔑 Neo-7B-Instruct 🔴 | 1126.2 | 36.3 | 39.5 | 14 | 15 | 31.4 | 25 | 1122.1 | 3735 |
| 48 | 🔑 StarlingLM-7B-beta | 1126.2 | 41.9 | 43.8 | 24.4 | 17 | 34.1 | 30.2 | 1126.3 | 2797 |
| 49 | 🔒 Command-R | 1125.6 | 44.1 | 47.4 | 19.3 | 16 | 34.6 | 29.5 | 1125.3 | 2919 |
| 50 | 🔑 Mixtral-8x7B-Instruct | 1124.7 | 41.9 | 42.8 | 25 | 22.1 | 34.6 | 31.5 | 1123.4 | 2653 |
| 51 | 🔑 Yi-1.5-6B-Chat | 1122.7 | 31.4 | 31.1 | 16.6 | 16.8 | 27.3 | 23.3 | 1110.3 | 3899 |
| 52 | 🔑 Tulu-2-dpo-70b | 1121 | 40.7 | 42.7 | 20.7 | 14.8 | 32.3 | 28 | 1119.1 | 2908 |
| 53 | 🔒 Reka-Edge | 1120.8 | 34.4 | 36.2 | 13.5 | 8.9 | 25 | 21.3 | 1112.2 | 2417 |
| 54 | 🔑 Mistral-7B-Instruct-v0.2 | 1105 | 40.1 | 42.1 | 18.4 | 10.1 | 30.1 | 25.6 | 1104.1 | 2832 |
| 55 | 🔑 Llama-2-70B-chat | 1101.9 | 38.3 | 40 | 9.3 | 4.2 | 26.8 | 20.7 | 1099.2 | 3138 |
| 56 | 🔑 Qwen1.5-7B-Chat | 1092.7 | 34 | 38.3 | 14.9 | 11.9 | 28.9 | 23.4 | 1091.1 | 2519 |
| 57 | 🔑 Hermes-2-Mixtral-8x7B-DPO | 1085.8 | 39.8 | 37.9 | 26 | 21.8 | 34.2 | 30.7 | 1083.6 | 2874 |
| 58 | 🔑 Phi-3-mini-128k | 1082.1 | 28.6 | 30.6 | 21.6 | 18.6 | 28.1 | 24.7 | 1074.5 | 2435 |
| 59 | 🔑 Gemma-7B-it | 1079.2 | 12.7 | 21.2 | 1.8 | -3.7 | 10.2 | 6.6 | 1054.5 | 1726 |
| 60 | 🔑 Llama-2-7B-chat | 1052.5 | 27.7 | 29.8 | -6.8 | -7.2 | 15.4 | 8.3 | 1044 | 2985 |
| 61 | 🔑 Gemma-2B-it | 1011.8 | -2.1 | 7.2 | -17.7 | -18.6 | -5.8 | -9.7 | 981.8 | 1590 |

Figure 7: Leaderboard of WildBench (2024 Sept 1st)