# OpenReview forum: "WildBench: Benchmarking LLMs with Challenging Tasks from Real Users in the Wild"
_ICLR.cc/2025/Conference — ICLR 2025 Spotlight_

### Official Review · Reviewer_vPGb · 2024-10-30

**Soundness:** 3
**Presentation:** 3
**Contribution:** 3
**Rating:** 8
**Confidence:** 5

**Summary:**

The authors introduced WildBench, a benchmarking framework for evaluating large language models (LLMs), using 1,024 complex tasks selected from real user conversation. These tasks reflect the natural distribution of real user tasks and are diverse, allowing for a more realistic evaluation of model capabilities. The framework incorporates detailed checklists to ensure objective evaluation. Experiments show that WildBench has a high correlation with human-voted Elo rankings from Chatbot Arena, outperforming other existing automated evaluation frameworks.

**Strengths:**

1. WildBench establishes a comprehensive evaluation pipeline, creating a task set from real user queries that is both well-distributed and diverse, enabling a more comprehensive evaluation. Additionally, WildBench can update over time to accommodate emerging task distributions, reducing some potential risks of data leakage.
2. Experimental results show that WildBench has a high correlation with the Elo scores from Chatbot Arena, which is considered a fair evaluation. However, the human cost is lower than Chatbot Arena, and WildBench provides two evaluation metrics, offering researchers diverse options for assessing LLM performance.
3. The evaluation incorporates task-specific checklists and considers length penalties, further reducing the negative impact of "LLM as a judge."

**Weaknesses:**

1. Although multi-turn conversation evaluation is a highlight of WildBench, the authors did not conduct much analysis from this perspective, which is unfortunate.
2. The task quality filtering method is not detailed enough or may have some inadequacies, which could lead to inaccurate evaluations of performance differences between specific models. Additionally, the analysis of the overall correlation with Chatbot Arena may not reflect the performance differences of specific models between the two leaderboards (e.g., the differences between Yi-Large and Yi-1.5-34B-Chat on WildBench and Chatbot Arena).
3. The test set includes multi-turn data, and the publicly available history conversation, when used for training models, still provides some assistance in answering the final question, which means there is still a certain risk of data leakage in the dataset.

**Questions:**

1. WildBench's feature of updating its task set over time is valuable. However, since the baseline models are fixed, have the authors considered whether the current baseline models can effectively evaluate models with updated training data and more advanced performance, as both the models and task sets evolve over a long period?
2. The responses in the history conversation come from earlier versions of ChatGPT-3.5 and GPT-4, which may introduce bias. For example, the history conversation may contain factual errors caused by hallucinations from ChatGPT-3.5 and GPT-4, which could conflict with the prior knowledge of the evaluated models, leading to inaccurate evaluations of model capabilities. Have any measures been taken to filter out such issues?
3. For multi-turn conversation filtering, besides limiting the number of turns and applying single-turn filtering rules, is there additional task theme consistency checking? Even with fewer turns, multiple themes might still be present, affecting task focus and coherence. There is also a potential issue where a task might consist of multiple relatively simple questions across different domains, making the overall difficulty appear higher and bypassing the difficulty filtering. However, the final question itself might be low in difficulty and easy to answer, affecting the overall benchmark difficulty.
4. The provided anonymous link is not accessible.

---

> ### Author Response · Authors · 2024-11-24
>
> Thank you for taking the time to provide a detailed review of our submission. We appreciate your constructive feedback and insightful suggestions, which help us further improve our work. Below, we address your concerns in detail.
>
> ---
>
> ### 1. Multi-Turn Conversation Evaluation
>
>    We appreciate your point about the lack of analysis from the multi-turn conversation perspective. We fully agree that more analysis is needed in this area, particularly regarding performance differences when varying the input context length. Given the limited time available during the review period, we plan to extend our evaluation to include a detailed analysis of multi-turn conversations in the camera-ready version. Specifically, we plan to investigate whether all models exhibit similar performance drops as the input context length changes. This will help provide a clearer understanding of how different models handle complex, multi-turn exchanges and allow us to identify any distinct patterns. We will incorporate this analysis in the final version to provide a more comprehensive understanding of WildBench's strengths in handling these types of queries.
>
> ---
>
> ### 2. Task Quality Filtering Method
>
>
>    Thank you for pointing out the potential shortcomings of the task quality filtering process. We plan to revisit the task selection and filtering mechanism to ensure a more robust evaluation in the camera-ready version. We also recognize that the correlation with Chatbot Arena might not fully capture the differences among specific models, particularly for variations like Yi-Large and Yi-1.5-34B-Chat. We plan to conduct further analysis to better represent the nuances of different models, and this information will be included in the final version.
>
> ---
>
> ### 3. Multi-Turn Data in Test Set
>
>
>    We understand the concern regarding the inclusion of publicly available conversation history, which could contribute to data leakage. This issue may be difficult to avoid entirely, as similar risks exist in other benchmarks like AlpacaEval and ArenaHard. Publicly released data always carries the potential for misuse in training, but we have taken measures to mitigate these risks. For instance, the data links are accompanied by consent forms requiring users to agree to specific usage guidelines. Additionally, we plan to introduce a private site for evaluation, where the inference code remains consistent, but the data is kept private. This will help ensure that models do not have access to public information, reducing the risk of data leakage.
>
> ---
>
> ### 4. Updating Task Set and Long-Term Evaluation
>
>
>    You raised a great point about the evolving performance of models with updated training data over time. To address this, we plan to investigate approaches to ensure that the benchmark remains adaptable and reflective of changes in LLM capabilities. We are already developing a follow-up version where we explore using O1 mini or even O1 preview for related evaluations and baselines. In our current submission, we considered this aspect by including three different baseline models to determine which parameterized evaluations are most effective and distinguishable. We will continue to retain this feature in future versions and enhance the task difficulty over time. Additionally, we will ensure that both the evaluator and baseline models are updated to stay current.
>
> ---
>
> ### 5. Hallucinations and Prior Knowledge in History Conversation
>
>    Thank you for bringing up the risk of hallucinations and conflicting information in the history conversations. To mitigate this issue, we have conducted a manual inspection to ensure that the final question in each multi-turn conversation is coherent and challenging. Our automatic data labeling process helps filter out unsuitable examples, but ultimately, every example is reviewed by experts to maintain quality. Moving forward, we will continue to enhance the filtering process to ensure that multi-turn questions are coherent, with the final question being a truly complex challenge that reflects the intended difficulty.
>
> ---
>
> ### 6. Anonymous Link Accessibility
>
>
>    Thank you for your note regarding the accessibility of the anonymous link. The anonymous link was intentionally kept restricted to protect the double-blind review policy. All links in the paper point to content that can be reproduced directly using the uploaded codebase. The website's content is consistent with the screenshots provided in the Appendix. Once the paper is accepted, we plan to replace the anonymous link with a publicly accessible version.
>
>
> ---
>
> ### Thank you!
>
> Thank you again for your thoughtful review and valuable suggestions. Your feedback is instrumental in refining our work, and we are confident that these revisions will lead to a stronger and more comprehensive submission. If you have any further questions or require additional clarifications, please feel free to reach out.

---

> > ### Comment · Reviewer_vPGb · 2024-11-26
> >
> > Thanks for your response. I have no further questions and will maintain my score.

---

### Official Review · Reviewer_EFf3 · 2024-11-02

**Soundness:** 3
**Presentation:** 3
**Contribution:** 3
**Rating:** 6
**Confidence:** 4

**Summary:**

This paper carefully picked out data from WildChat to construct an evaluation benchmark with in-the-wild user queries.

This paper proposes two kinds of evaluation strategies, namely, checklists and structured analysis, to encourage more detailed and fine-grained evaluation.

This paper proposes two primary metrics for both pairwise and pointwise evaluation and further mitigates the bias towards longer outputs.

**Strengths:**

- The first benchmark for LLM evaluation with in-the-wild user queries.
- A bunch of tricks for a more accurate and debiased evaluation:
    - pairwise + pointwise
    - checklist-based
    - multiple baseline models for pairwise evaluation.
    - structured analysis before concluding the rewards.
    - mitigating the length bias with a margin term $K$ for ties, which is customizable according to the users’ needs.
- Interesting experimental findings such as the re-evaluated performance of LLaMA3-Inst-8B-SimPO.

**Weaknesses:**

- The novelty of this work is quite limited despite its richness of evaluation and wild data construction practices.
- The handling of length penalty in this work is a bit crude, because sometimes a longer response is indeed better. Performing uniform processing directly at the entire dataset level fails to take into account the specific situations of each sample.
- The experimental findings offer limited novelty compared to existing benchmarks like AlpacaEval and Chatbot Arena. The comparative performance among different models remains largely consistent with previous evaluations. Additionally, this work lacks sufficient analysis from the perspective of "wildness" in the data. It fails to address a crucial question: What unique insights emerge when introducing "wildness" into the evaluation process?

**Questions:**

In the weakness part.

---

> ### Author Response · Authors · 2024-11-24
> **Author Response (1/2)**
>
> Thank you for liking our paper and giving thoughtful reviews, as well as recognizing our strengths. We appreciate your detailed review of our submission and your valuable suggestions, which have helped us identify areas for improvement. Below, we address your concerns in detail.
>
> ---
>
>
> ### 1. Novelty of the Work
>
> **Respectfully, we disagree with the assessment that our contributions are limited in novelty.** While we do use large language models as evaluators, similar to other benchmarks such as AlpacaEval, we have introduced several significant innovations that set our work apart.
>
> **First, we leverage a large corpus of real-world user data to identify challenging tasks for authentic evaluations.** **Unlike previous benchmarks, our focus on difficult, real-user queries provides a realistic assessment of LLM capabilities in handling complex and diverse situations.** For example, many questions in AlpacaEval are quite simple and do not effectively differentiate model capabilities. Questions such as "What is the capital of Australia?" or repetitive recipe-related inquiries (e.g., over 20 similar recipe questions) do not challenge the model's abilities adequately. In contrast, our approach in WildBench aims to evaluate models with more complex tasks, thus providing a better understanding of model robustness.
>
> Second, regarding automated evaluation, we adopt a better approach compared to others like AlpacaEval. **Instead of simply choosing between response A or B without providing reasons, we use a Chain of Thought (CoT) methodology, where the evaluator analyzes each response step by step, noting strengths and weaknesses before arriving at a final decision.** This detailed analysis adds transparency and depth to the evaluation process, which is not commonly employed on such a large scale in previous benchmarks.
>
> **Moreover, we use a checklist to guide the evaluation process, which ensures the focus is on the model's problem-solving ability rather than overfitting to GPT preferences.** This also helps evaluators pay closer attention to different facets of the responses, making the evaluation more holistic.
>
> **Finally, we incorporate multiple baseline models to make pairwise evaluations more accurate and reliable, ensuring robustness across different model comparisons.** The checklist plays a crucial role in our evaluation process by ensuring that evaluators systematically consider multiple aspects of a response, such as relevance, coherence, and factual accuracy. This structured approach helps avoid overfitting to GPT preferences and focuses on a comprehensive assessment of problem-solving abilities. By using a checklist, we provide a more reliable and thorough evaluation, which ultimately enhances the robustness and fairness of our benchmark.
>
> ---
>
> ### 2. Handling of Length Penalty
>
> **We appreciate your feedback on the handling of length penalties.** We would like to clarify that our current approach only applies length penalties under specific conditions: when the response length exceeds a certain threshold, and when the evaluation outcome is "slightly better" rather than "much better" or "slightly worse." **We agree that this approach may not always be entirely accurate, as longer responses can sometimes be preferable.** Therefore, we allow users to adjust the length penalty parameter (K) to create a personalized evaluation. If a user prefers longer or shorter responses, they can modify K accordingly, providing flexibility for different evaluation preferences.
>
> **In future versions, we also plan to implement more comprehensive solutions for handling length penalties.** For example, during annotation, we intend to have our evaluators label multiple aspects of a response, such as style, factuality, and helpfulness. This approach will help us better differentiate stylistic elements, especially advantages related to response length, from other important evaluation dimensions.
>
> (to be continued)

---

> ### Author Response · Authors · 2024-11-24
> **Author Response (2/2)**
>
> ### 3. Compared to Existing Benchmarks
>
> **Our benchmark focuses on the "wildness" of user interactions, which we believe is a significant differentiator.** Our evaluation results, based on real user queries, demonstrate a closer alignment with human-based evaluations, such as those seen in ChatBot Arena. Unlike ChatBot Arena, which relies on human evaluators and lacks the scalability needed for rapid assessment of developing models, **WildBench provides an automated yet reliable evaluation framework that mirrors human preferences.** This also allows us to generate synthetic feedback efficiently, which is not possible with purely human-based evaluation methods.
>
> We also acknowledge your point regarding the results involving SimPO. In many cases, models or methods might overfit to certain superficial features, which can deceive GPT-based evaluators, especially in simpler methods like AlpacaEval that do not use our checklist and CoT-based judgment. For example, SimPO or similar methods and models might achieve very high scores on AlpacaEval (sometimes an 8B model can even outperform GPT-4), yet they fail to replicate similar results on WildBench. **This difference highlights how our data and evaluation methods better align with real user experiences, avoiding overfitting to GPT-4 preferences for simple questions, and instead focusing on improving the true helpfulness of the model.** This is the effect we aim to achieve with WildBench.
>
> ---
>
> ### 4. Beyond a Dataset and Evaluation Work
>
> **We believe that WildBench is more than just an evaluation framework; it is also a tool for modeling synthetic feedback for natural and challenging tasks from real users.** **One of our major contributions is demonstrating that WildBench evaluations are strongly correlated with human judgments, meaning our automated method can effectively simulate human preferences at scale with an affordable cost.** This capability opens up opportunities for generating large-scale data to support subsequent RLHF (Reinforcement Learning from Human Feedback) and DPO (Direct Preference Optimization) training processes. **This aspect also reflects the "wildness" of our approach, emphasizing its applicability to real-world, complex scenarios.**
>
> ---
>
> ## Thank You!
>
> Thank you again for your thoughtful and valuable comments. Your feedback has provided us with an opportunity to further refine our work, and we are confident that these revisions have strengthened our submission. Please feel free to reach out if you have any further questions or need additional clarifications. We kindly hope that you would please consider raising the score for our submission if you feel that we have adequately addressed your concerns. We would greatly appreciate it, and we are committed to incorporating these responses into the final version of the paper.

---

> ### Comment · Reviewer_EFf3 · 2024-11-27
> **Reviewer Response**
>
> A1: I acknowledge the first novelty. But for the remaining two novelties, there are numerous works that leverage explanations and checklists to enhance the LLM judge [1][2].
>
> [1] CheckEval: Robust Evaluation Framework using Large Language Model via Checklist
>
> [2] TIGERScore: Towards Building Explainable Metric for All Text Generation Tasks
>
> ---
>
> A2: Thanks for your clarifications. I am looking forward to more fine-grained aspects.
>
> ---
>
> A3: Thanks for your valuable experimental findings.
>
> ---
>
> Thanks again for your kind clarifications. I am willing to keep my positive score.

---

### Official Review · Reviewer_LYdL · 2024-11-03

**Soundness:** 3
**Presentation:** 3
**Contribution:** 4
**Rating:** 8
**Confidence:** 4

**Summary:**

The paper introduces WILDBENCH, a novel benchmarking framework designed to evaluate large language models (LLMs) using challenging, real-world user queries. It consists of 1,024 examples selected from over one million human-chatbot conversation logs. The framework employs two evaluation metrics, WB-Reward and WB-Score, to provide reliable and interpretable assessments of LLM performance. By using task-specific checklists and a dynamic update mechanism, WILDBENCH aims to reflect real user interactions and mitigate data leakage concerns. The results demonstrate a strong correlation with human judgment, surpassing existing benchmarks and offering a more accurate reflection of LLM capabilities.

**Strengths:**

1. **Real-World Evaluation**: utilizes real user queries to provide a more reliable assessment of LLM performance. This approach is more cost-effective than human evaluations and offers greater authenticity compared to traditional machine evaluations.

2. **Robust Evaluation Metrics**: introduces WB-Reward and WB-Score, which enhance accuracy through the use of checklists and a CoT-style evaluation process. These metrics effectively mitigate length bias, ensuring fairer comparisons.

3. **Dynamic Updates and Data Leakage Prevention**: update regularly, reflecting new user interactions and preventing data leakage, which enhances the benchmark's relevance and security.

**Weaknesses:**

1. **Cost Comparison**: The paper does not provide a detailed comparison of the llm api costs between WILDBENCH and previous evaluation methods, which could help in understanding its efficiency.

2. **Cheaper Version with Consistency Check**: Offer a cheaper version of the benchmark, such as using 30% of the samples, and verify it's the consistency and reliability with the full evaluation results.

**Questions:**

See weakness part.

---

> ### Author Response · Authors · 2024-11-24
> **Response to Reviewer LYdL**
>
> Thank you for recognizing the value of our work, particularly its real-world evaluation, robust metrics, and dynamic updates. We greatly appreciate your feedback, and below we provide detailed responses to your comments and suggestions.
>
> ---
>
> ### Cost Comparison
>
> Thank you for your suggestion to provide a more detailed cost comparison between WILDBENCH and previous evaluation methods. Using the official codebase from AlpacaEval and ArenaHard, the evaluation cost is approximately \$10 and \$40, respectively. In contrast, WildBench's codebase supports batch mode API evaluation, resulting in a more efficient cost of around \$13-\$17.
>
> ---
>
> ### Lite Version with Consistency Check
>
> Thank you for the suggestion regarding a more cost-efficient version of our benchmark. We have been developing a lite version using 25% of the hardest subset. The selection criteria leverage existing evaluation results from over 70 models to identify examples with lower average scores and higher variance, ensuring the subset remains representative. For this new subset, we are considering using o1-mini for evaluation, which costs around \$5-\$7 per evaluation, while maintaining high correlation with the full evaluation and accurately reflecting differences across models.&#x20;
>
> We plan to host a new leaderboard with this subset soon, encouraging broader use of our evaluation framework while making it more accessible and affordable. We will continue to provide support for the evaluation of any models that are openly accessible.
>
> ---
>
> ### Thank You!
>
> We sincerely appreciate your time and effort in providing these insightful suggestions. Your feedback has been instrumental in improving both the clarity and practical aspects of our benchmark. We value the opportunity to address your concerns and are confident that the updates we have made strengthen our submission. If there are any further questions or points needing clarification, we would be more than happy to address them.

---

### Meta-Review · Area_Chair_LVxi · 2024-12-19

**Metareview:**

This paper introduces WildBench, an evaluation benchmark comprising test data filtered from the WildChat data, an automatic evaluation protocol, and evaluation metrics. The data source corresponds to real interactions between humans and chatbots. The evaluation data is carefully curated using a mix of heuristics (e.g., length-based or relying on meta-data such as topic and user ids) and assisted filtering with language models to ensure relevance and diversity. The filtering process results in 1024 test examples that somewhat uniformly covers topics/tasks, unlike some popular existing benchmarks that are somewhat skewed towards certain tasks. An automated evaluation is proposed where three language models are employed independently and combined so as to smooth away the evaluation variance. Results are observed to correlate strongly with human performance. The paper is clearly written and easy to follow, and reviewers are unanimously in favor of acceptance.

While I side with reviewers, appreciate the contribution and think the community will greatly benefit from it, and recommend acceptance, there are some limitations worth noting nonetheless:

- Risk of contamination: although authors highlight they reserve a subset of WildChat that is never released publicly, since the dataset is itself a result of user interactions with deployed language models, that data may have been used for training of the models involved in these interactions.
- Cost of evaluation: Evaluation relies on running inference with multiple language models, and estimates on cost (both time and financial cost) to eval are not thoroughly discussed. Given that open source models are improving fast and catching up with proprietary models, it would have been great to see to what extent an eval using only open-source models correlate with the proposal in the paper. It's also worth noting that the evaluation imposes geographical constraints since models used for evaluation may not be available everywhere.

**Additional Comments On Reviewer Discussion:**

Reviewers were mostly satisfied with the clarifications provided by the authors.

---

### Decision · Program_Chairs · 2025-01-22

Accept (Spotlight)